# The use of methylprednisolone in COVID-19 patients: A propensity score matched retrospective cohort study

Xiang You[1☯], Chao-hui Wu[1☯], Ya-nan Fu[2], Zonglin He[3,4], Pin-fang Huang[1], Gong-ping Chen[5], Cui-hong Lin[1], Wai-kit Ming[3,4‡]*, Rong-fang Lin[1‡]*

**1** Department of Pharmacy, The First Affiliated Hospital of Fujian Medical University, Fuzhou, China, **2** Department of Pharmacy, Yichang Third People's Hospital, Yichang, China, **3** Department of Public Health and Preventive Medicine, School of Medicine, Jinan University, Guangzhou, China, **4** Faculty of Medicine, International School, Jinan University, Guangzhou, China, **5** Respiratory Department, The First Affiliated Hospital of Fujian Medical University, Fuzhou, China

☯ These authors contributed equally to this work.
‡ WM and RL also contributed equally to this work.
* wkming@connect.hku.hk (WM); Fsnowing@126.com (RL)

**Data Availability Statement:** All relevant data are within the manuscript and its Supporting information files.

## Abstract

### Purpose

To evaluate the efficacy and safety of methylprednisolone in treating the coronavirus disease 2019 (COVID-19) patients.

### Methods

A retrospective cohort study was conducted, and all COVID-19 patients were recruited who were admitted to the Yichang Third People's Hospital from February 1st to March 31st, 2020. One-to-one propensity score matching (PSM) was used for minimizing confounding effects. The primary outcome was hospital mortality, with the secondary outcomes being the time needed for a positive SARS-CoV-2 nucleic acid test to turn negative and the length of hospital stay.

### Results

Totaling 367 patients with COVID-19 hospitalized at the Yichang Third People's Hospital were identified, of whom 276 were mild or stable COVID-19, and 67 were serious or critically ill. Among them, 255 patients were treated using methylprednisolone, and 188 did not receive any corticosteroid-related treatment. After PSM, no statistically significant difference was found in the baseline characteristics between the two groups. Regarding the outcomes, there also were no statistically significant difference between the two groups. Patients without the use of methylprednisolone were more quickly to obtain negative results of their nasopharyngeal swab tests of SARS-CoV-2 nucleic acid after treatment, compared to those receiving methylprednisolone.

**Funding:** The study was supported by the Social Development Guiding (Key) Project of Fujian Provincial Science and Technology Department, 2020Y0027, Pin-fang Huang.

**Competing interests:** The authors have declared that no competing interests exist.

## Conclusion

Methylprednisolone could not improve the prognosis of patients with COVID-19, and the efficacy and safety of the use of methylprednisolone in patients with COVID-19 still remain uncertain, thus the use of corticosteroids clinically in patients with COVID-19 should be with cautions.

## Introduction

An unprecedented novel pneumonia-like disease outbreak named the 2019 Coronavirus disease (COVID-19), which was caused by a novel coronavirus coined the severe acute respiratory syndrome (SARS)-CoV-2, has swept China and wreaked havoc across the globe [1]. As of mid-June, 215 countries or regions have reported confirmed cases, with more than 6,800,000 confirmed cases and 400,000 deaths, and a death rate over 5.84% worldwide [2].

Patients with COVID-19 showed primary symptoms similar to those with SARS and Middle East respiratory syndrome (MERS), such as fever, dry cough, myalgia, and fatigue [3]. In addition, severe pneumonia caused by the coronavirus can increase the levels of inflammatory cytokines, resulting in acute lung injury (ALI) and acute respiratory distress syndrome (ARDS) [4–6]. In 2003, systemic methylprednisolone was widely used to treat patients infected with the SARS virus because of its anti-inflammatory effect [7], although potential risks have been found when used to treat coronavirus-related diseases [8–10]. Until now, there has been no clear evidence that COVID-19 patients can benefit from treatment with methylprednisolone. Therefore, despite of the recommendation of use of methylprednisolone put forward by the seventh version of *New Coronavirus Pneumonia Prevention and Control Guideline* published by the National Health Commission of China, the use of methylprednisolone remains controversial [11]. Hence it is necessary to scrutinize the effects of methylprednisolone on the outcomes of COVID-19 patients.

With the report of the effectiveness of dexamethasone in reducing death in hospitalized patients with COVID-19 by RECOVERY trial, how its analogous substitute, methylprednisolone, can exert effects on patients with COVID-19 were of interest [12]. Therefore, the objectives of this retrospective cohort study were as follows: (i) to describe the clinical characteristics of 343 patients with COVID-19, (ii) to compare the clinical characteristics of patients receiving methylprednisolone or not, and (iii) to assess the efficacy and safety of the use of methylprednisolone in patients with COVID-19. The findings from this study may shed light into the effectiveness of the use of corticosteroids in patients with COVID-19 and may help enhance patient care in the future. Specifically, to reduce the confounding effects, propensity score matching has been used.

## Methods

### Study design and cohort description

A retrospective, nonrandomized intervention study was conducted by reviewing the clinical records of all patients with confirmed COVID-19 hospitalized in the Yichang Third People's Hospital from February 1st to March 31st, 2020. The design of this study was reviewed and approved by the Ethics Committee of the First Affiliated Hospital of Fujian Medical University (No. MRCTA, ECFAH of FMU [2020]153), and the consent of the patients was waived by the committee owing to the retrospective nature of the present study and the data being analyzed

anonymously. The study inclusion criteria were as follows: (1) laboratory confirmed COVID-19; (2) age $>$ 18 years. The exclusion criteria were: (1) the patients whose records with missed or incomplete information; (2) the patients whose nucleic acid test has turned negative prior to the treatment; (3) the patients who were still hospitalized after March 31[st], 2020.

The diagnostic criteria of COVID-19 used were based on the *New Coronavirus Pneumonia Prevention and Control Guideline*, *version 7*, published by the National Health Commission of *China* [11]. A confirmed case of COVID-19 infection was defined as the satisfaction of the following conditions: (1) existence of SARS-CoV-2 nucleic acid detected by RT-PCR; (2) gene sequencing showed that the virus gene was highly homologous to any known novel coronaviruses; (3) tests for novel coronavirus IgM and IgG in the patient's blood serum were positive; or a test for IgG turned positive or its value was 4-fold higher in the acute phase.

## Data collection

All clinical information, including epidemiological, clinical, laboratory, and radiological characteristics, together with data regarding the treatment, outcomes, and complications, were collected from the electronic medical record databases in the hospital from February 1[st] to March 31[st], 2020 by two researchers independently. The data were reviewed and scrutinized by a third clinician, and the difference was adjudicated by discussion.

## Outcomes

The primary outcomes were hospital mortality. Secondary outcomes included the time needed for a positive nucleic acid test to turn negative, the length of hospital stay. the number of patients who required oxygen inhalation, occurrence of decreased oxygenation saturation, and any complications including shock, ARDS, acute kidney injury and liver damage.

## Definitions

The disease classification was defined according to the *New Coronavirus Pneumonia Prevention and Control Guideline* (*version 7*) *for COVID-19*. A positive-turned-negative nucleic acid test was defined as two successive negative results for nucleic acid tests that were performed with a 24-hour interval between the tests. The severity of COVID-19 was defined according to the diagnostic and treatment guidelines for SARS-CoV-2 issued by the Chinese National Health Committee version 7. Oxygen inhalation was defined as the use of a nasal catheter oxygen inhalation device or a non-invasive ventilator. Acute respiratory distress syndrome and shock were diagnosed by physicians clinically.

Severe COVID-19 was designated as having one of the following criteria: (i) respiratory distress with respiratory frequency $\geq$30/min; (ii) pulse oximeter oxygen saturation $\leq$93% at rest; and (iii) oxygenation index (artery partial pressure of oxygen/inspired oxygen fraction, PaO2/ FiO2) $\leq$300 mm Hg.

The duration from disease onset to hospital admission, disease progression and aggravation, and discharge were also recorded. All patients who received corticosteroids only received methylprednisolone only in this study, and the patients receiving methylprednisolone therapy would receive methylprednisolone 40 mg once or twice per day within 48 hours of admission for one week.

## Patient and public involvement

Patients and public were not involved in the study.

## Statistical analysis

Propensity scores matching (PSM) is useful in reducing selection biases by matching exposed and unexposed patients based on the baseline covariates [13]. This approach identifies neighborhood that are identical to each other with respect to the probability of being in the exposed group. The following PSM algorithm is used in the present study:

$$1/N1 * \text{Var}\,(Y \mid \text{Methylprednisolone} = 1) + \text{Sum}\,(w\_i^2; i \text{ in Methylprednisolone}$$
$$= 0)/(N1)^2 * \text{Var}(Y \mid \text{Methylprednisolone} = 0)$$

where N1 is the number of matched treated, Methylprednisolone = 1 denotes the patients receiving the methylprednisolone therapy, Methylprednisolone = 0 denotes the patients that did not receive the methylprednisolone therapy, and w_i is the weight given to control i.

In the present study, in view of the potential bias and the possible exaggerated effects of the observed groups secondary to the increased matching ratio, we chose the one-to-one PSM instead of one-to-two or one-to-many scheme [14]. After one-to-one PSM algorithm, with the confounding variables being age, sex, history of tobacco smoking, history of alcohol drinking, and presence of comorties including hypertension, diabetes, coronary heart diseases, cerebral cardiovarscular diseases, chronic obstructive pulmonary disease, malignancy, hepatitis, and the severity of COVID-19 assessed upon admission to the hospital, a total of 255 patients were ruled out [15]. Therefore, the remaining total cohort (n = 88) were then labeled as the Matched Methylprednisolone Group (n = 44) and the Matched Non-methylprednisolone Group (n = 44), whose confounding effects were minimized, and the difference between two groups were statistically insignificant.

All statistical analyses were conducted using software Stata MP 14.0 (Stata corp., USA). Normally distributed continuous variables were expressed as a mean and standard deviation (SD), while skewed continuous variables were expressed as median and interquartile range (IQR), and values for categorical variables are expressed as frequency counts and percentages.

Means for normally distributed continuous variables were compared using independent sample t-tests, and the Kruskal-Wallis test was used to analyze continuous variables that did not follow the normal distribution. Proportions for categorical variables were compared using the Chi-squared test, where those with limited counts were compared using the Fisher's exact test. Differences were considered significant for $P$-values < 0.05.

## Results

### Patient demographics and baseline characteristics

In the present study, a total of 367 patients with COVID-19 being hospitalized at the Yichang Third People's Hospital were identified. And after excluding 8 patients who were < 18 years old, 12 patients whose nucleic acid test had turned negative after therapy at other hospitals, and 4 patients with missing data, finally a total of 343 patients were included in our final analysis, as shown in Fig 1. The demographics and baseline characteristics of the enrolled patients are presented in Table 1. Among the 343 patients, 276 of them were classified as mild or stable COVID-19, while 67 were designated as serious or critically ill COVID-19. Moreover, 255 of them were treated using methylprednisolone and 188 did not receive any corticosteroid-related treatment and underwent usual care alone.

The mean age of patients receiving methylprednisolone was 56.75 years old (SD = 15.41 years), around 6 years older than that of their counterparts, those who did not receive methylprednisolone (mean = 50.89 years, SD = 17.46 years) (P <0.01, Table 1). Males accounted for

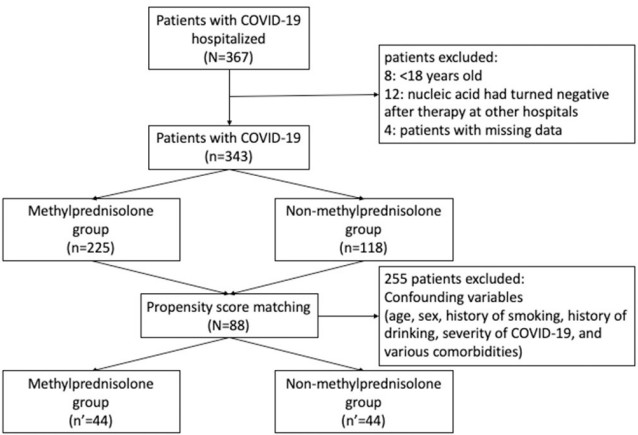

**Fig 1. The flow chart of subject selection.**

66 (55.9%) and 120 (53.3%) of the cases of patients receiving methylprednisolone and those not receiving it, respectively (P <0.01).

Around 90.7% of patients receiving methylprednisolone exhibited fever as symptoms, 81.0% of which were shown as initial symptoms, while there were only 70.7% exhibiting fever in the control group. Moreover, patients receiving methylprednisolone were more likely to exhibit classic COVID-19 symptoms, viz., dry cough [95 (82.6%) vs. 155 (68.9%)], expectoration [69 (58.5%) vs. 85 (37.8%)], shortness of breath [36 (30.5%) vs. 8 (3.6%)], headache [7 (5.9%) vs. 7 (3.1%)], and fatigue [61 (51.7%) vs. (59 (26.2%))] than their counterparts. More patients receiving methylprednisolone would be classified as seriously or critically ill at admission.

Owing to the high heterogeneity of the baseline characteristics, we performed the PSM with as many significantly different baseline characteristics variables as confounders to minimize the confounding effects, thus we can better discern the genuine effects of the use of methyl-prednisolone in patients with COVID-19. Therefore, after PSM, no statistically significant difference could be discerned in the baseline characteristics between two groups (Table 1). And still, patients with methylprednisolone tended more to exhibit fever [39(89%) vs. 33(75%), P = 0.097], especially as initial symptoms [38 (86%) vs. 26(60%), P<0.05].

## Treatment and clinical outcomes

Various therapies have been implemented, apart from the use of methylprednisolone, including antiviral treatment, Chinese patent medicine, immune enhancers, secretolytic agent, antibacterial treatment, and antifungal therapy, as shown in Table 2. After PSM, in terms of antiviral treatment, oseltamivir [40 (91%) vs. 32 (73%)], as well as lopinavir and ritonavir [39 (89%) vs. 27 (61%)] were used more in patients receiving methylprednisolone than in patients did not use methylprednisolone; nevertheless, arbidol [22 (50%) vs. 14 (32%)] was more used in patient who did not use methylprednisolone. Moreover, 9 (20%) of cases in patient with the use of methylprednisolone received gamma-immunoglobulin, while only 1 (2%) patient without using methylprednisolone used it.

Moreover, the effects of the use of prednisolone in critically ill patient with COVID-19 were assessed, and the results revealed that prednisolone played a limited role in rescuing these patients, as shown in S1 and S2 Tables.

**Table 1. Baseline characteristics of patients with COVID-19 receiving methylprednisolone and non-methylprednisolone therapy.**

| | Unmatched | | | Propensity Score Matched | | |
|---|---|---|---|---|---|---|
| | Non-methylprednisolone group | Methylprednisolone group | p-value | Non-methylprednisolone group | Methylprednisolone group | p-value |
| | (N = 225) | (N = 118) | | (N = 44) | (N = 44) | |
| Age, mean (SD) | 50.89 (17.46) | 56.75 (15.41) | 0.002 | 56.82 (17.32) | 54.25 (15.28) | 0.46 |
| Age group | | | 0.003 | | | 0.59 |
| <45 years old | 93 (41.3%) | 31 (26.3%) | | 13 (30%) | 13 (30%) | |
| 45–65 years old | 82 (36.4%) | 42 (35.6%) | | 15 (34%) | 19 (43%) | |
| >65 years old | 50 (22.2%) | 45 (38.1%) | | 16 (36%) | 12 (27%) | |
| Sex | | | 0.65 | | | 0.51 |
| Female | 105 (46.7%) | 52 (44.1%) | | 29 (66%) | 26 (59%) | |
| Male | 120 (53.3%) | 66 (55.9%) | | 15 (34%) | 18 (41%) | |
| Presence of Comorbidities | | | | | | |
| History of tobacco smoking | 36 (16.0%) | 22 (18.8%) | 0.51 | 6 (14%) | 8 (18%) | 0.56 |
| History of alcohol consumption | 19 (8.4%) | 5 (4.2%) | 0.15 | 1 (2%) | 3 (7%) | 0.31 |
| Hypertension | 42 (18.7%) | 39 (33.1%) | 0.003 | 14 (32%) | 14 (32%) | 1 |
| Diabetes | 20 (8.9%) | 22 (18.6%) | 0.009 | 5 (11%) | 4 (9%) | 0.72 |
| Malignancy | 4 (1.8%) | 5 (4.2%) | 0.18 | 0 (0%) | 0 (0%) | / |
| Hepatitis B | 10 (6.6%) | 4 (5.2%) | 0.67 | 1 (2%) | 2 (5%) | 0.56 |
| History of being in Wuhan | 75 (35.9%) | 53 (47.7%) | 0.039 | 13 (32%) | 16 (40%) | 0.49 |
| History of ever contracting with patients infected with COVID-19 | 56 (47.5%) | 17 (29.3%) | 0.022 | 12 (57%) | 7 (33%) | 0.12 |
| Symptoms and signs | | | | | | |
| Fever | 159 (70.7%) | 107 (90.7%) | <0.001 | 33 (75%) | 39 (89%) | 0.097 |
| Fever as initial symptoms | 138 (61.6%) | 94 (81.0%) | <0.001 | 26 (60%) | 38 (86%) | 0.006 |
| Maximum body temperature (˚C) | | | <0.001 | | | 0.05 |
| • <37.3 | 71 (31.6%) | 10 (8.5%) | | 14 (32%) | 5 (11%) | |
| • 37.3–38.0 | 74 (32.9%) | 25 (21.2%) | | 11 (25%) | 11 (25%) | |
| • 38.1–39.0 | 76 (33.8%) | 61 (51.7%) | | 19 (43%) | 24 (55%) | |
| • >39.0 | 4 (1.8%) | 17 (14.4%) | | 0 (0%) | 3 (7%) | |
| • Missing values | 0 (0.0%) | 1 (0.8%) | | 0 (0%) | 1 (2%) | |
| Dry cough | 155 (68.9%) | 95 (82.6%) | 0.007 | 21 (48%) | 25 (57%) | 0.39 |
| Expectoration (coughing phlegm) | 85 (37.8%) | 69 (58.5%) | <0.001 | 0 (0%) | 0 (0%) | / |
| Shortness of breath | 8 (3.6%) | 36 (30.5%) | <0.001 | 3 (7%) | 7 (16%) | 0.18 |
| Headache | 7 (3.1%) | 7 (5.9%) | 0.21 | 0 (0%) | 0 (0%) | / |
| Fatigue | 59 (26.2%) | 61 (51.7%) | <0.001 | 14 (32%) | 24 (55%) | 0.031 |
| Diarrhea | 10 (4.4%) | 3 (2.5%) | 0.38 | 2 (5%) | 3 (7%) | 0.65 |
| Time from illness to first hospital admission (Days), median (IQR) | 3 (2, 5) | 4 (3, 7) | 0.011 | 3 (2, 5) | 3 (2, 7) | 0.81 |
| Severity assessment at admission | | | <0.001 | | | 0.47 |
| Mild | 6 (2.7%) | 0 (0.0%) | | 1 (2%) | 0 (0%) | |
| Stable | 210 (93.3%) | 60 (50.8%) | | 37 (84%) | 38 (86%) | |
| Serious | 7 (3.1%) | 26 (22.0%) | | 5 (11%) | 3 (7%) | |
| Critical | 2 (0.9%) | 32 (27.1%) | | 1 (2%) | 3 (7%) | |

The patients received a mean dosage of 260.85 mg (SD = 175.07 mg) prednisolone during their stay in hospital, in which the methylprednisolone therapy underwent an average of 5 day (IQR = 3,6). In terms of the outcomes, there were no statistically significant difference between the two groups. And interestingly, patients without the use of methylprednisolone were more

**Table 2. Treatment and outcomes of patients with COVID-19 receiving methylprednisolone and non-methylprednisolone therapy.**

| | Unmatched | | | Propensity score matched | | |
|---|---|---|---|---|---|---|
| | Non-methylprednisolone group | Methylprednisolone group | p-value | Non-methylprednisolone group | Methylprednisolone group | p-value |
| | (N = 225) | (N = 118) | | (N = 44) | (N = 44) | |
| Treatment | | | | | | |
| Antiviral Treatment | 225 (100.0%) | 117 (99.2%) | 0.17 | 44 (100%) | 44 (100%) | / |
| Interferon | 153 (68.0%) | 63 (53.8%) | 0.010 | 35 (80%) | 35 (80%) | 1.00 |
| Oseltamivir | 165 (73.3%) | 92 (78.0%) | 0.35 | 32 (73%) | 40 (91%) | 0.027 |
| Arbidol | 111 (49.3%) | 55 (46.6%) | 0.63 | 22 (50%) | 14 (32%) | 0.083 |
| Lopinavir and ritonavir | 140 (62.2%) | 96 (81.4%) | <0.001 | 27 (61%) | 39 (89%) | 0.003 |
| Ribavirin | 30 (13.3%) | 11 (9.3%) | 0.28 | 6 (14%) | 3 (7%) | 0.29 |
| Ganciclovir | 5 (2.2%) | 5 (4.2%) | 0.29 | 0 (0%) | 1 (2%) | 0.31 |
| Chinese patent medicine | | | | | | |
| Xuebijing | 106 (47.1%) | 78 (66.1%) | <0.001 | 25 (57%) | 26 (59%) | 0.83 |
| Tanreqing | 48 (21.4%) | 35 (29.7%) | 0.091 | 10 (23%) | 14 (32%) | 0.34 |
| Reduning | 76 (33.9%) | 32 (27.1%) | 0.20 | 10 (23%) | 8 (18%) | 0.60 |
| Immune enhancers | | | | | | |
| Thymalfasin | 29 (12.9%) | 36 (30.5%) | <0.001 | 5 (11%) | 5 (11%) | 1.00 |
| Gamma-immunoglobulin | 11 (4.9%) | 54 (45.8%) | <0.001 | 1 (2%) | 9 (20%) | 0.007 |
| Ambroxol | 23 (10.2%) | 33 (28.0%) | <0.001 | 3 (7%) | 3 (7%) | 1.00 |
| Antibacterial treatment | 206 (91.6%) | 113 (95.8%) | 0.15 | 42 (95%) | 41 (93%) | 0.65 |
| Antifungal treatment | 3 (1.3%) | 17 (14.4%) | <0.001 | 2 (5%) | 0 (0%) | 0.15 |
| Duration of methylprednisolone therapy (Days), median (IQR) | 0.00 (0.00, 0.00) | 5.00 (3.00, 8.00) | <0.001 | 0.00 (0.00, 0.00) | 5.00 (3.00, 6.00) | <0.001 |
| Drug amount of methylprednisolone used, mean (SD) | 0.00 (0.00) | 369.37 (314.58) | <0.001 | 0.00 (0.00) | 260.85 (175.07) | <0.001 |
| Outcomes | | | | | | |
| Outcomes | | | | | | |
| Discharge | 224 (99.6%) | 103 (87.3%) | <0.001 | 43 (98%) | 44 (100%) | 0.31 |
| Death | 1 (0.4%) | 14 (11.9%) | | 1 (2%) | 0 (0%) | |
| Missing values | 0 (0.0%) | 1 (0.8%) | | | | |
| Aggravation of illness | | | | | | |
| No change | 218 (96.9%) | 74 (62.7%) | <0.001 | 40 (91%) | 37 (84%) | 0.53 |
| Stable -> Serious | 5 (2.2%) | 14 (11.9%) | | 3 (7%) | 4 (9%) | |
| Serious -> critical | 2 (0.9%) | 30 (25.4%) | | 1 (2%) | 3 (7%) | |
| Length of hospital stay (Days), median (IQR) | 21 (17, 27) | 27.5 (20, 35) | <0.001 | 22.5 (19.5, 29) | 23.5 (20, 32) | 0.21 |
| Time taken for Nasopharyngeal swab tests results to turn negative(days), median (IQR) | 11 (9, 15) | 13.5 (10, 19) | 0.018 | 10 (7, 16) | 11 (9, 15.5) | 0.40 |
| Oxygen support | | | | | | |
| Nasal catheter | 159 (70.7%) | 64 (54.2%) | <0.001 | 35 (80%) | 30 (68%) | 0.38 |
| High-flow oxygen or non-invasive | 5 (2.2%) | 32 (27.1%) | | 2 (5%) | 5 (11%) | |
| Invasive | 0 (0.0%) | 12 (10.2%) | | 0 (0.0%) | 0 (0.0%) | |
| No need | 61 (27.1%) | 10 (8.5%) | | 7 (16%) | 9 (20%) | |
| Oxygen saturation ≤93% | 12 (5.3%) | 35 (29.7%) | <0.001 | 4 (9%) | 8 (18%) | 0.21 |
| Acute kidney injury | 3 (1.3%) | 7 (5.9%) | 0.016 | 2 (5%) | 0 (0%) | 0.15 |
| Acute respiratory distress syndrome | 3 (1.3%) | 21 (17.8%) | <0.001 | 1 (2%) | 3 (7%) | 0.31 |
| Use of reduced glutathione | 33 (14.7%) | 49 (41.5%) | <0.001 | 9 (20%) | 16 (36%) | 0.098 |
| Liver protection therapy | 33 (14.8%) | 47 (39.8%) | <0.001 | 10 (23%) | 14 (32%) | 0.34 |

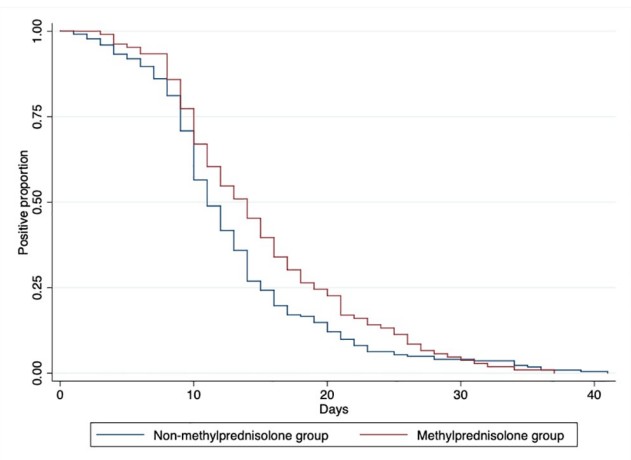

**Fig 2. Nasopharyngeal swab tests of SARS-CoV-2 nucleic acid by RT-PCR between the two groups.**

quickly to obtain negative results of their nasopharyngeal swab tests of SARS-CoV-2 nucleic acid after treatment, compared to those receiving methylprednisolone (Fig 2).

## Discussion

In this retrospective cohort study, using PSM, we found that methylprednisolone did not exert positive effects on reducing adverse outcomes of patients with COVID-19, or reducing the incidence of ARDS; and in critically ill patients with COVID-19, patients receiving methyl-prednisolone were even associated with a much higher death rate. Patients did not require methylprednisolone as a routine treatment drug, because it did not reduce the length of their hospital stay or time of nucleic acid turning negative. However, we found that the length of stay hospital and number of patients who required oxygenation therapy increased with patient age. These findings may suggest that only a small number of the patients could be benefited from taking methylprednisolone.

Zha et al. [16] conducted a study in which corticosteroids were used to reduce the risk of ARDS in patients with viral infections; however, their study did not classify patients according to the severity of illness, and enrolled fewer patients (n = 31) than our study (n = 276). In our study, around 33% of the patients had comorbidities, and 67 of the patients were critically ill. In critically ill patients, the disease itself can affect the patient's clinical manifestations. It is worth noting that the results of the two studies (our study and Zha et al.) were consistent in terms of the time needed for nucleic acid turning negative and the length of hospital stay [16].

Some studies suggested that cytokine storms and viral evasion of cellular immune responses play important roles in determining disease severity [17, 18]. The generation of a cytokine storm can lead to ARDS, which is a leading cause of death among patients with severe acute respiratory syndrome [6, 19]. Methylprednisolone was widely used to treat SARS and Middle East respiratory syndrome. However, in this study we did not observe a significant increase in cases where patients developed ARDS in the non-methylprednisolone group, and there may be several reasons for that result. First, the patients included in our study were not critically ill and did not show signs of a cytokine storm. Second, the immune regulation function of the patients was normal, which may have prevented a cytokine storm. Third, there might have been a decrease in virulence when compared with the early stage of the COVIS-19 outbreak.

In general, the pathogenesis of this new coronavirus is still not completely understood, and the blind use of methylprednisolone is not worth promoting.

## Strengths and limitations

Our study has some limitations. First, the dosages of methylprednisolone could only be estimated, because the weight of the patients could not be accurately measured as most of the patients used a stretcher or wheelchair. Second, the single-center retrospective nature of the study allows patients in the study to have received a wide variety of other pharmaceutical and herbal therapy, and the limited number of the subjects included in the study has undermined the statistical power. An important strength of our study is the utilization of PSM algorithm to reduce the selection bias, and sampling bias. By measuring and adjusting for all known and measurable confounding variables, this approach yields efficient estimates for observational studies; nevertheless, we cannot exclude some bias due to residual confounding [20].

Therefore, we urged larger-scale, multiple-centered, and prospective studies to be conducted to further verify our results and elaborate on the mechanisms underlying the use of steroids in patients with COVID-19.

## Conclusion

Methylprednisolone did not shorten a patient's hospital stay or reduce the likelihood that their disease would worsen, and this was especially true for patients greater than 65 years old. Patients treated with methylprednisolone were more likely to require oxygen inhalation. The efficacy and safety of the use of methylprednisolone in patients with COVID-19 still remain uncertain.

## Supporting information

**S1 Table. The base line characteristics of the critically ill and non-critically ill COVID-19 patients.**
(DOCX)

**S2 Table. Treatment and outcomes of critically ill and non-critically ill patients with COVID-19 receiving methylprednisolone and non-methylprednisolone therapy.**
(DOCX)

## Acknowledgments

We sincerely thank the Yichang Third People's Hospital for providing patient data.

## Author Contributions

**Conceptualization:** Xiang You, Chao-hui Wu, Ya-nan Fu, Pin-fang Huang.

**Data curation:** Xiang You, Chao-hui Wu, Ya-nan Fu.

**Formal analysis:** Zonglin He.

**Methodology:** Zonglin He, Gong-ping Chen.

**Project administration:** Cui-hong Lin, Wai-kit Ming, Rong-fang Lin.

**Resources:** Pin-fang Huang.

**Supervision:** Gong-ping Chen, Cui-hong Lin, Wai-kit Ming, Rong-fang Lin.

**Visualization:** Zonglin He.

**Writing – original draft:** Xiang You, Chao-hui Wu, Ya-nan Fu, Zonglin He, Pin-fang Huang, Gong-ping Chen, Cui-hong Lin, Wai-kit Ming, Rong-fang Lin.

**Writing – review & editing:** Zonglin He, Wai-kit Ming, Rong-fang Lin.

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
