## [Decision Letter · Decision Letter 0]

26 Nov 2020

PONE-D-20-33050

The use of methylprednisolone in COVID-19 patients: a propensity score matched retrospective cohort study

PLOS ONE

Dear Dr. Ming,

Thank you for submitting your manuscript to PLOS ONE. After careful consideration, we feel that it has merit but does not fully meet PLOS ONE’s publication criteria as it currently stands. Therefore, we invite you to submit a revised version of the manuscript that addresses the points raised during the review process.

The reviewers have commented on your above paper. They have suggested that this manuscript be revised according to the reviewers suggestions and resubmitted.  Provided you address the changes recommended, the manuscript will be accepted for publication.

We look forward to receiving your revised manuscript.

Kind regards,

Prof. Raffaele Serra, M.D., Ph.D

Academic Editor

PLOS ONE

Additional Editor Comments:

The reviewers have commented on your above paper. They have suggested that this manuscript be revised according to the reviewers suggestions and resubmitted.

Journal Requirements:

Reviewers' comments:

Reviewer's Responses to Questions

**Comments to the Author**

1. Is the manuscript technically sound, and do the data support the conclusions?

Reviewer #1: Yes

2. Has the statistical analysis been performed appropriately and rigorously? 

Reviewer #1: Yes

3. Have the authors made all data underlying the findings in their manuscript fully available?

Reviewer #1: Yes

4. Is the manuscript presented in an intelligible fashion and written in standard English?

Reviewer #1: No

5. Review Comments to the Author

Reviewer #1: The authors are to be congratulated on their thorough and careful analysis of the safety and efficacy of methylprednisolone therapy for patients with COVID-19 infection. Their single center, retrospective, analysis of patient outcomes is enhanced by the use of propensity score matching. The authors demonstrate that administration of methylprednisolone for patients with SARS-CoV-2 infection did not significantly improve patient outcomes. The authors also demonstrate that patients who did not receive methylprednisolone therapy converted more quickly to negative SARS-CoV-2 nasopharyngeal PCR test results than did patients who received methylprednisolone therapy. Weaknesses of their study include the following: this is a single center review, patients in the study received a wide variety of other pharmaceutical and herbal therapy, and only 44 propensity score matched patients in each of the two groups were available for analysis. The strengths of their study include: the completeness of the data they make available, their rigorous statistical analysis, and the significance of their findings. This manuscript requires English language editing.

6. PLOS authors have the option to publish the peer review history of their article (what does this mean?). If published, this will include your full peer review and any attached files.

Reviewer #1: No

---

## [Author Response · Author response to Decision Letter 0]

3 Dec 2020

Prof. Raffaele Serra

RE: Manuscript ID PONE-D-20-33050 

Dear Prof. Raffaele Serra,

Thank you very much for your letter and advice, and thank the reviewer for the positive comments of our manuscript. We have revised the paper and would like to re-submit it for your consideration. We have carefully read all the comments and addressed all issues raised by the reviewer as follows. All the changes made within the manuscript are highlighted and in blue color. I look forward to hearing from you soon. 

Yours sincerely,

Zonglin He

 

Response to reviewer: 

Reviewer #1: 

The authors are to be congratulated on their thorough and careful analysis of the safety and efficacy of methylprednisolone therapy for patients with COVID-19 infection. Their single center, retrospective, analysis of patient outcomes is enhanced by the use of propensity score matching. The authors demonstrate that administration of methylprednisolone for patients with SARS-CoV-2 infection did not significantly improve patient outcomes. The authors also demonstrate that patients who did not receive methylprednisolone therapy converted more quickly to negative SARS-CoV-2 nasopharyngeal PCR test results than did patients who received methylprednisolone therapy. 

Weaknesses of their study include the following: this is a single center review, patients in the study received a wide variety of other pharmaceutical and herbal therapy, and only 44 propensity score matched patients in each of the two groups were available for analysis. The strengths of their study include: the completeness of the data they make available, their rigorous statistical analysis, and the significance of their findings. This manuscript requires English language editing.

Response: We thank the reviewer for the positive comment of our manuscript. And we have revised the manuscript accordingly as follows: 

(line 93 page 5) “A retrospective, nonrandomized intervention study was conducted by reviewing the clinical records of all patients with confirmed COVID-19 hospitalized in the Yichang Third People’s Hospital from February 1st to March 31st, 2020.”

(line 100 page 5) “The exclusion criteria were: (1) the patients whose records with missed or incomplete information; (2) the patients whose nucleic acid test has turned negative prior to the treatment; (3) the patients who were still hospitalized after March 31st, 2020. ”

(line 205 page 14) “Males accounted for 66 (55.9%) and 120 (53.3%) of the cases of patients receiving methylprednisolone and those not receiving it, respectively (P <0.01). 

Around 90.7% of patients receiving methylprednisolone exhibited fever as symptoms, 81.0% of which were shown as initial symptoms, while there were only 70.7% exhibiting fever in the control group. Moreover, patients receiving methylprednisolone were more likely to exhibit classic COVID-19 symptoms, viz., dry cough [95 (82.6%) vs. 155 (68.9%)], expectoration [69 (58.5%) vs. 85 (37.8%)], shortness of breath [36 (30.5%) vs. 8 (3.6%)], headache [7 (5.9%) vs. 7 (3.1%)], and fatigue [61 (51.7%) vs. (59 (26.2%)) than their counterparts. More patients receiving methylprednisolone would be classified as seriously or critically ill at admission. ”

(line 282 page 20) “Strengths and limitations

Our study has some limitations. First, the dosages of methylprednisolone could only be estimated, because the weight of the patients could not be accurately measured as most of the patients used a stretcher or wheelchair. Second, the single-center retrospective nature of the study allows patients in the study to have received a wide variety of other pharmaceutical and herbal therapy, and the limited number of the subjects included in the study has undermined the statistical power. An important strength of our study is the utilization of PSM algorithm to reduce the selection bias, and sampling bias. By measuring and adjusting for all known and measurable confounding variables, this approach yields efficient estimates for observational studies; nevertheless, we cannot exclude some bias due to residual confounding [21]. ”

---

## [Editor Report · Decision Letter 1]

4 Dec 2020

The use of methylprednisolone in COVID-19 patients: a propensity score matched retrospective cohort study

PONE-D-20-33050R1

Dear Dr. Ming,

We’re pleased to inform you that your manuscript has been judged scientifically suitable for publication and will be formally accepted for publication once it meets all outstanding technical requirements.

Kind regards,

Prof. Raffaele Serra, M.D., Ph.D

Academic Editor

PLOS ONE

Additional Editor Comments (optional):

amendend manuscritp is acceptable.
---

## [Editor Report · Acceptance letter]

23 Dec 2020

PONE-D-20-33050R1 

The use of methylprednisolone in COVID-19 patients: a propensity score matched retrospective cohort study 

Dear Dr. Ming:

I'm pleased to inform you that your manuscript has been deemed suitable for publication in PLOS ONE. Congratulations! Your manuscript is now with our production department. 

Kind regards, 

on behalf of

Prof. Raffaele Serra 

Academic Editor

PLOS ONE